# Hyperparameter Selection in Continual Learning

## Abstract

In continual learning (CL)—where a learner trains on a stream of data—standard hyperparameter optimisation (HPO) cannot be applied, as a learner does not have access to all of the data at the same time. This has prompted the development of CL-specific HPO frameworks. The most popular way to tune hyperparameters in CL is to repeatedly train over the whole data stream with different hyperparameter settings. However, this *end-of-training* HPO is unrealistic as in practice a learner can only see the stream once. Hence, there is an open question: *what HPO framework should a practitioner use for a CL problem in reality?* This paper answers this question by evaluating several realistic HPO frameworks. We find that all the HPO frameworks considered, including end-of-training HPO, perform similarly on common CL benchmarks. We therefore advocate using the realistic and most computationally efficient method: fitting the hyperparameters on the first task and then fixing them throughout training.

## 1 Introduction

Sequentially updating deep learning systems on a non-stationary data stream is a challenging problem which *continual learning* (CL) methods aim to address. The standard CL setup is when a learner sees a sequence of tasks one-by-one and at the end of learning is evaluated on how well it performs across all tasks. There have been many methods (Delange et al., 2021; Parisi et al., 2019; Wang et al., 2023) designed for this problem and CL scenarios proposed (Hsu et al., 2018; Antoniou et al., 2020; van de Ven & Tolias, 2019). A key decision when using a CL method is selecting hyperparameter settings—learning rates, regularisation coefficients, etc. (Feurer & Hutter, 2019; Delange et al., 2021; Wistuba et al., 2023). The most common way to fit hyperparameters for CL is *end-of-training* hyperparameter optimisation (HPO) (Delange et al., 2021; Buzzega et al., 2020)—shown in Figure 1. This is when the hyperparameters are fit by training over the whole data stream with each hyperparameter configuration and then selecting the configuration that has the best end-of-training performance on a held-out validation set. However, end-of-training HPO is unrealistic as in the real world a learner can only train over the data stream once and must select hyperparameters only using the data it can currently access. Therefore, it is currently unclear what is the best *realistic* way to perform HPO for CL.

In this work we address the problem of deciding what HPO framework to use in CL. We benchmark a variety of approaches for performing HPO in CL across different CL methodologies (ER (Chaudhry et al., 2020), ER-ACE (Caccia et al., 2021), iCaRL (Rebuffi et al., 2017), ESMER (Sarfraz et al., 2023) and DER++ (Buzzega et al., 2020)). We investigate both fixed HPO frameworks where the hyperparameters are kept constant throughout training and dynamic HPO frameworks where hyperparameters are adapted throughout learning. For fixed HPO we examine (i) *end-of-training* HPO as well as (ii) a *first-task* HPO framework where we fit the hyperparameters only using data from the first task (see Figure 1), a realistic and computationally efficient method. For dynamic HPO, we consider (i) using data from the current task, (ii) using data stored in memory, and (iii) using validation sets from previous tasks to perform HPO for each new task. By comparing these different HPO frameworks we shed light on what validation signal is sufficient to fit hyperparameters in CL and whether hyperparameters need to be adapted during training.

Our experiments show that all the HPO frameworks tested perform similarly in terms of predictive performance; no one method is consistently better than the others. This could be because in standard CL

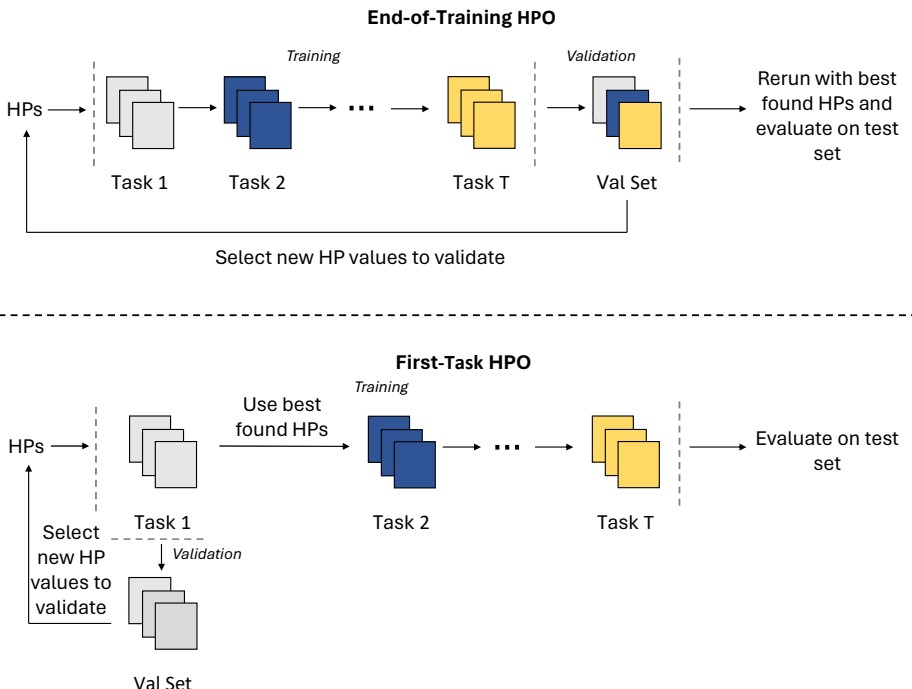

Figure 1: Depiction of end-of-training and first-task HPO frameworks, which fix the hyperparameters (HPs) throughout training. *End-of-training* HPO is the most common HPO framework for CL and works by training over the whole data stream for each HP configuration and then uses a validation set consisting of data from each task to select the best HPs. End-of-training HPO is unrealistic as it assumes you have access to all of the data stream from the start of training. While, *first-task* HPO selects HPs by repeatedly training and validating performance on the first task, which is more realistic and efficient.

benchmarks each task is very homogeneous, being of the same size and difficulty. Hence, there might be no reason to change hyperparameters per task or to use more than the first task to select hyperparameters. We therefore also evaluate the performance of each HPO framework on more heterogeneous data streams where the tasks are of varying difficulty and have different numbers of data points associated with them. However, even in this case all HPO frameworks perform similarly. Given these results we advocate the realistic and most computationally efficient method, *first-task* HPO, to practitioners for performing HPO in CL.

The main contributions of this work are:

- We benchmark a suite of realistic CL HPO frameworks against the commonly used but unrealistic end-of-training HPO.

- We show that all HPO frameworks we compare perform similarly in our experiments, including both methods which dynamically change hyperparameters throughout training and those which do not.

- We provide evidence on common CL benchmarks that first-task HPO is a good method for performing HPO, as it performs similarly to other approaches while being more computationally efficient.

## 2 Preliminaries and related work

CL is a large research area where many different settings have been looked at. In this work we look at the most common CL setting which is known as standard CL, or sometimes offline CL (Prabhu et al., 2020). In *Standard CL*, the learner sees a non-stationary sequence of data chunks called *tasks* one-by-one, such that it only has access to one chunk at a time and cannot access previously seen or future chunks. Each task

consists of examples which are data instance and label pairs (e.g. pairs of images and their class) sampled from a subset of the classes. For example, the first task might be examples of cows and sheep and the second task could be formed of examples of dogs and cats. The goal of the learner is to classify new examples accurately after training on the whole data stream. There are two common ways to evaluate a CL learner, task and class incremental learning. *Task-incremental* learning is when, at test-time, the learner knows which task a data instance comes from and so only needs to distinguish between classes within that task. While, *class-incremental* learning is when the learner is not given what task a data instance belongs to at test time and must distinguish between all classes from all the tasks. An important part of the standard CL setting is the assumption of memory constraints, which is why a learner cannot solve CL by storing previous data chunks in memory. The memory constraints take the form of only allowing a learner to store a small amount of previous data in memory and in constraining its use of memory for storing additional networks or parts of networks (Delange et al., 2021; Wang et al., 2023).

There have been many methods proposed for CL (Delange et al., 2021; Parisi et al., 2019; Wang et al., 2023). One of the most popular and performant approaches to standard CL are replay methods (Wang et al., 2023). This is especially true for class-incremental learning, where they are commonly the best performing methods (van de Ven & Tolias, 2019; Wu et al., 2022; Mirzadeh et al., 2020; Lee & Storkey, 2024). *Replay* methods use a memory buffer to store a set of examples from previous tasks to regularise the updates on new tasks such that the learner does not forget previous task knowledge. For example, the stereotypical replay method is *experience replay* (ER) (Chaudhry et al., 2020; 2019b; Aljundi et al., 2019a) which for each learning step appends a sample of data from the replay buffer to the batch of current task data to be trained on. More complex replay methods often use a form of knowledge distillation on a sample of data from the replay buffer. For example, DER++ (Buzzega et al., 2020), ESMER (Sarfraz et al., 2023) and iCaRL (Rebuffi et al., 2017) are replay methods which use a method-specific knowledge distillation term. For each of these methods the most common hyperparameters that are tuned are the learning rate and regularisation coefficients, which need to be tuned to get good performance (see Appendix B). While other potential hyperparameters are often not tuned in CL, e.g. momentum (Buzzega et al., 2020).

While the most common HPO framework used in standard CL is end-of-training HPO, there have been several other HPO frameworks suggested (Kilickaya & Vanschoren, 2023; Parisi et al., 2019). For example, Delange et al. (2021) propose a dynamic HPO framework. The method adapts the hyperparameters for each task by first training with the hyperparameter configuration which is assumed to have the least impact on previous task performance. Then the method incrementally changes hyperparameter values to improve performance on the current task to a prespecified value, while decreasing performance on previous tasks. However, this method assumes that the direction to change hyperparameters to increase performance on the current task is known and that the interaction between different hyperparameters is understood. In this work we look at a similar HPO framework, current-task HPO, which does not need the above assumptions. Also, for the online CL scenario—which is different to standard CL—another HPO framework has been proposed whereby end-of-training HPO is used on the first (or first $k$) tasks and then the hyperparameters are fixed after that (Chaudhry et al., 2019a). To the best of our knowledge, this HPO framework has been rarely used in standard CL up to this point. Here, we look at it in the form of the first-task HPO framework and examine how it performs in the commonly used standard CL setting. There has also been work on making dynamic HPO frameworks more efficient by sampling fewer HPO configurations, for example using bandit methods (Liu et al., 2023) and analysis of variance techniques (Semola et al., 2024). However, for simplicity, we only look at the more expensive dynamic HPO frameworks which are an upper bound to the performance of these more efficient methods. In this work we have aimed to have tested the main proposed HPO frameworks to see which performs the best and note that to the best of our knowledge no previous works have comprehensively compared the different HPO frameworks for CL to each other.

## 3 Standard CL

While the setting we look at, standard CL, is mentioned above, we describe it more formally here. In *standard CL* a learner sees a sequence of tasks, $D_1, \ldots, D_T$, where each task consists of a chunk of data. The chunks of data consist of a set of examples, where an example is a pair formed of a data instance $\mathbf{x} \in X$ and label $y \in C$. Each task only contains examples from a given subset of the classes, in other words for all

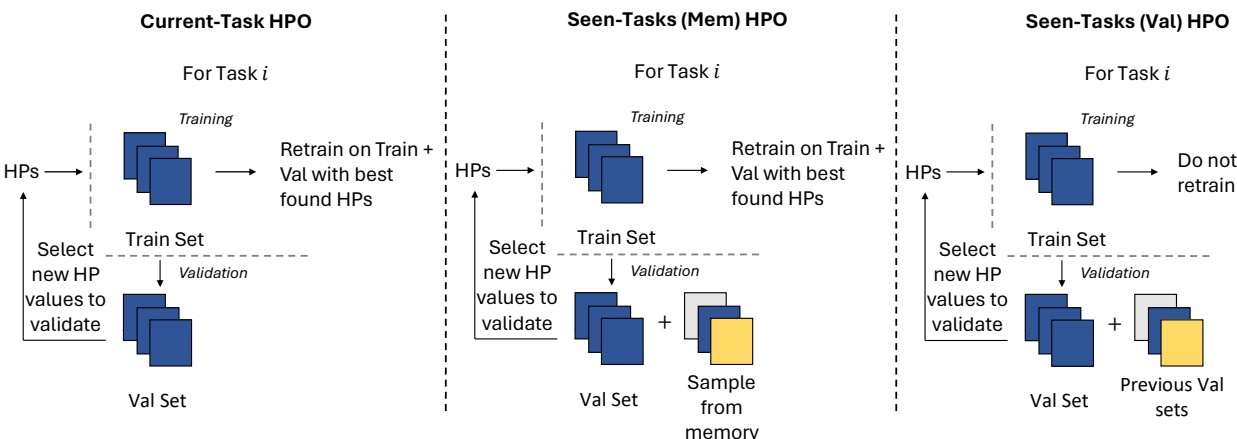

Figure 2: Depiction of current-task, seen-tasks (Mem) and seen-tasks (Val) HPO frameworks, which dynamically adapt hyperparameters (HPs) for each task. Each methods splits the data of the current task into train and validation sets. Then, current-task HPO uses this validation set to fit the HPs for the current task. While, seen-tasks (Mem) and seen-tasks (Val) use a combination of this validation set and either a sample of data from previous tasks stored in memory or validation sets of previous tasks, respectively. Then current-task and seen-tasks (Mem) HPO retrain on the combined validation and train sets to complete the learning process on that task. Seen-tasks (Val) does not retrain, instead it takes the model fitted using the best found hyperparameters as the final model for the current task. This is to ensure that the current task's validation set has not been trained on when fitting hyperparameters for future tasks.

$(\mathbf{x}, y) \in D_i$ we have that $y \in C_i$ and $C_i \subseteq C$ is the subset of classes the examples of that task can belong to. In this work we look at the most common setting, where no two tasks have examples from the same class. This means that for any two task $i$ and $j$ we have that $C_i \cap C_j = \varnothing$. Additionally, learners can have a memory buffer of previous examples which consists at task $i$ of the set $M_i$. Training consists of the learner sequentially seeing each task in order and it cannot access the data from previous or future tasks. For each task, its data chunk is split into training and validations sets, $\text{Train}_i \subseteq D_i$ and $\text{Val}_i \subseteq D_i$, to enable the use of HPO frameworks. Then after fitting the hyperparameters the learner usually retrains on the combination of the training and validation sets, $D_i = \text{Train}_i \cup \text{Val}_i$. After training the learner is tested by evaluating its performance on a held-out set of data which consists of an equal number of examples from all the classes. We look at two evaluation scenarios, task-incremental learning and class-incremental learning. *Task-incremental* learning is where the learner receives with each test data instance the task it belongs to and therefore the subset of classes that the data instance can belong to. While for *class-incremental* learning, no indication is given of what task a test data instance belongs to.

## 4 HPO frameworks for CL

In this work, we examine several HPO frameworks for CL to see which should be the preferred choice to use in CL. We look both at fixed HPO frameworks which keep the values of hyperparameters constant throughout training and dynamic HPO which adapts the hyperparameters per task. The fixed HPO frameworks we look at are end-of-training HPO and first-task HPO and the dynamic HPO frameworks we look at are current-task HPO, seen-tasks HPO (Mem) and seen-tasks HPO (Val). Each of these frameworks are described in turn below and we present an overview of their advantages and disadvantages in Table 1.

**End-of-training HPO** is the most common HPO framework for CL (shown in Figure 1). It selects hyperparameters by first training each hyperparameter configuration on the *whole data stream*. Second, it evaluates the final model fitted using each hyperparameter configuration on a validation set formed of each task's held-out validation set, and selects the configuration with the highest validation performance. Last, it retrains using the selected configuration on the whole data stream where the validation data for each task is

Table 1: Advantages and disadvantages of different HPO frameworks. Where, for time complexity, $K$ refers to the number of hyperparameter configurations looked at and $T$ is the number of tasks in the data stream. The asterisk (*) for seen-tasks HPO (Val) denotes that, while it does not require knowledge of future tasks like end-of-training HPO, it does require additional storage compared to other methods. The additional memory is needed to store the validation sets of previous tasks.

| HPO Framework | Realistic? | Efficient? (Time Complexity) |
|---|---|---|
| End-of-training HPO | ✗ | ✗ ($\mathcal{O}(T \times K)$) |
| First-task HPO | ✓ | ✓ ($\mathcal{O}(T + K)$) |
| Current-task HPO | ✓ | ✗ ($\mathcal{O}(T \times K)$) |
| Seen-tasks HPO (Val) | ✓* | ✗ ($\mathcal{O}(T \times K)$) |
| Seen-tasks HPO (Mem) | ✓ | ✗ ($\mathcal{O}(T \times K)$) |

added to the training data. The model fitted at the end of this training run is the final model to be evaluated. This HPO framework is expensive as it needs to perform a training run over all the data stream for each hyperparameter configuration looked at. Additionally, it is unrealistic as it requires running through the data stream multiple times, which is not possible in many real-world settings. It might be thought that to make end-of-training more realistic the learner could store a network for each hyperparameter configuration: updating each network on every task and performing selection at the end of training. This idea would remove the requirement of running through the data stream multiple times. However, it would also require a large amount of extra memory, linear in the number of hyperparameter configurations. Additionally, the learner would have to store and not train on the validation data for each previous task. Therefore, because of underlying constraints on memory usage in standard CL, it is not realistic to use such an idea, and if there was enough memory available it would probably be better spent in other areas such as storing more data from previous tasks.

**First-task HPO** is a fixed HPO framework which is illustrated in Figure 1. It selects hyperparameters by training each hyperparameter configuration on the *first task*. Next, it measures the performance of each configuration on the held-out validation set of the first task. The configuration with the highest validation accuracy is then used to retrain on the first task using both the training and validation data and thereafter for all of the future tasks. First-task HPO is computationally efficient as it trains using each hyperparameter configuration solely on the first task and then only trains using one configuration for the rest of the tasks. This is much less costly than end-of-training HPO, which for all tasks must train using each hyperparameter configuration. Additionally, first-task HPO can be used in real-world settings as it only assumes access to data available at the start of training, the first task, and not future tasks like end-of-training HPO.

**Current-task HPO** is a dynamic HPO framework which selects hyperparameters for each task using the validation set of the *current task* (shown in Figure 2). This is a greedy strategy, selecting the hyperparameters that maximise the validation performance of the current task. It is roughly as computationally expensive as end-of-training HPO, as it has to validate each hyperparameter configuration for each task. However, it is more realistic than end-of-training HPO as it only needs access to the current task's data.

**Seen-tasks HPO (Mem) and seen-tasks HPO (Val)** are dynamic HPO frameworks (shown in Figure 2). They select hyperparameters for each task by a validation set formed of current task validation data along with some historic data from the stream. We consider two ways to integrate historic task data. Seen-tasks HPO (Mem) uses a sample of data from the current memory buffer. Seen-tasks HPO (Val) uses the validation sets of previous tasks. So, unlike current-task HPO, the hyperparameters are fit using both current and previous task data. This should aid the HPO procedure in selecting hyperparameters that ensure previous tasks are not forgotten. Like current task HPO, both seen-tasks HPO (Mem) and seen-tasks HPO (Val) are as computationally expensive as end-of-training HPO. Seen-tasks HPO (Val) assumes it is possible to access the validation sets of previous tasks which makes it less realistic than current or first task HPO. This is unlike seen-tasks HPO (Mem) which does not assume this as it uses data stored in the memory buffer to measure performance on the previous tasks. But, this comes at the cost of biasing its validation performance as the data in the memory buffer has been trained on in previous tasks.

For seen-tasks HPO (Mem), three additional details are important to mention. First, to ensure we are not training on validation data, the sample from memory used in the validation set is not trained on for the current task. Second, as the memory buffer contains different amounts of data for each task, we sample the same proportion of examples from each task to add to the validation set. Last, unlike for the other HPO frameworks, the validation set combined with the sample from memory might be class imbalanced. Therefore, unlike other methods which use validation accuracy as the performance metric, for seen-tasks HPO (Mem) we use the median of per class accuracies to reduce the impact of class imbalance.

## 5 Experiments

**Benchmarks** In our experiments we look at two settings, the commonly used split task setting (Buzzega et al., 2020; Delange et al., 2021) and the heterogeneous task setting. We look at these settings using the datasets CIFAR-10, CIFAR-100 and Tiny ImageNet (Krizhevsky, 2009; Wu et al., 2015). In the split task setting, each task has the same number of classes associated with it and no two tasks share a class. For CIFAR-10, the dataset is split into five tasks, each containing the data from two of the classes. For CIFAR-100 and Tiny ImageNet, the datasets are split into ten tasks, where each task contains the data of 10 or 20 classes, respectively. In the heterogeneous task setting, instead of each task having the same number of classes associated with it they have a varying amount, from two to ten, but still no two tasks share a class (see Appendix A for more details). This is to make the tasks have differing amounts of data and difficulty. We only look at CIFAR-100 and Tiny ImageNet for the heterogeneous task setting as CIFAR-10 contains a small number of classes, making it impossible to vary the number of classes by a large degree. Additionally, for the heterogeneous task setting we divide the datasets into twenty tasks to test how HPO frameworks perform on longer task sequences. For both settings, if required by the HPO framework, we split the data of the task into train and validations sets, where the validation set contains 10% of the task's data evenly sampled from each class associated with the task.

We evaluate the methods at the end of training using a standard performance metric for CL, average accuracy (Chaudhry et al., 2019a). The average accuracy of a method is the mean accuracy over each task on a held-out test set which contains an equal amount of data from each task. For class-incremental learning, the learner must classify between all classes at test time as it is not told what task a test data instance comes from. For task-incremental learning, the learner knows what task each test data instance comes from, meaning only classes from that task will be predicted.

**CL methods** To evaluate how well each HPO framework performs we look at applying them to fit the hyperparameters of several common and well performing CL methods. More specifically, we utilise the CL methods: ER (Chaudhry et al., 2020), ER-ACE (Caccia et al., 2021), iCaRL (Rebuffi et al., 2017), ESMER (Sarfraz et al., 2023) and DER++ (Buzzega et al., 2020). For these methods we fit the learning rate and any regularisation coefficients they have using each HPO framework. While all HPO frameworks looked at can be used with any underlying sampler/selector of hyperparameter configurations, for simplicity and to be consistent with common practice in CL (Buzzega et al., 2020; Boschini et al., 2022; Sarfraz et al., 2023) we use grid search. We look at the combination of ten different learning rate values and for each regularisation coefficient three different values. This means for DER++ we search over 90 different hyperparameter configurations (learning rate and two regularisation coefficients) and for ESMER we search over 30 different configurations (learning rate and the loss margin coefficient). While, for ER, iCaRL and ER-ACE we look at 10 different configurations as they have no regularisation coefficients to fit. The hyperparameter grid used is very similar to the ones looked at in several popular works on CL (Buzzega et al., 2020; Boschini et al., 2022) and is given in full in Appendix A. Moreover, for each method we use: a ResNet18 (He et al., 2016) as the underlying backbone network; random crop and horizontal flip data augmentations when training; and a memory buffer of size 5120, as in common with previous work (Buzzega et al., 2020).

### 5.1 Results

For the split task setting, the results of our experiments show that none of the HPO frameworks looked at perform much better than the rest. The results are presented in Table 2 and we have bolded the results which are better by +0.5% than any of the other HPO frameworks results for a given CL method. The reason we

Table 2: Results of using different HPO frameworks for ER, iCaRL, ER-ACE, ESMER and DER++ on standard CL benchmarks, i.e. the split task setting. We report mean average accuracy over three runs with their standard errors and bold results which are greater by +0.5% average accuracy than any other for that CL method. The table shows that all HPO frameworks perform similarly and that the simplest and least computationally expensive method, first-task HPO, performs as well as the rest.

| CL Method | HPO Framework | CIFAR-10 | | CIFAR-100 | | TinyImageNet | |
| | | Class-IL. | Task-IL. | Class-IL. | Task-IL. | Class-IL. | Task-IL. |
| --- | --- | --- | --- | --- | --- | --- | --- |
| ER | End-of-training HPO | $83.55_{\pm0.44}$ | $97.18_{\pm0.14}$ | $51.03_{\pm0.43}$ | $85.68_{\pm0.29}$ | $28.01_{\pm0.09}$ | $68.17_{\pm0.06}$ |
| | First-task HPO | $\mathbf{84.38}_{\pm0.45}$ | $96.82_{\pm0.17}$ | $49.61_{\pm0.34}$ | $84.97_{\pm0.19}$ | $28.51_{\pm0.18}$ | $\mathbf{68.72}_{\pm0.13}$ |
| | Current-task HPO | $82.10_{\pm2.21}$ | $96.39_{\pm0.50}$ | $50.64_{\pm0.40}$ | $85.47_{\pm0.18}$ | $25.79_{\pm0.21}$ | $66.96_{\pm0.15}$ |
| | Seen-tasks HPO (Val) | $83.67_{\pm0.73}$ | $96.84_{\pm0.21}$ | $51.46_{\pm0.36}$ | $85.65_{\pm0.06}$ | $28.45_{\pm0.28}$ | $68.16_{\pm0.26}$ |
| | Seen-tasks HPO (Mem) | $79.49_{\pm0.63}$ | $95.93_{\pm0.09}$ | $47.39_{\pm0.24}$ | $84.83_{\pm0.22}$ | $\mathbf{29.58}_{\pm0.25}$ | $68.02_{\pm0.14}$ |
| iCaRL | End-of-training HPO | $77.79_{\pm0.23}$ | $\mathbf{98.52}_{\pm0.03}$ | $54.30_{\pm0.36}$ | $85.74_{\pm0.45}$ | $37.09_{\pm0.27}$ | $70.37_{\pm0.36}$ |
| | First-task HPO | $77.83_{\pm0.22}$ | $95.31_{\pm0.12}$ | $52.56_{\pm0.10}$ | $84.60_{\pm0.09}$ | $36.42_{\pm0.22}$ | $70.11_{\pm0.13}$ |
| | Current-task HPO | $76.15_{\pm0.75}$ | $93.29_{\pm0.61}$ | $54.26_{\pm0.02}$ | $85.74_{\pm0.06}$ | $37.17_{\pm0.28}$ | $70.67_{\pm0.03}$ |
| | Seen-tasks HPO (Val) | $77.58_{\pm0.49}$ | $94.32_{\pm1.01}$ | $51.89_{\pm0.39}$ | $84.02_{\pm0.68}$ | $34.81_{\pm0.42}$ | $68.42_{\pm0.41}$ |
| | Seen-tasks HPO (Mem) | $76.67_{\pm0.44}$ | $95.41_{\pm0.28}$ | $49.16_{\pm0.23}$ | $82.43_{\pm0.23}$ | $36.79_{\pm0.13}$ | $70.46_{\pm0.08}$ |
| ER-ACE | End-of-training HPO | $82.34_{\pm0.30}$ | $96.74_{\pm0.01}$ | $55.58_{\pm0.39}$ | $85.73_{\pm0.09}$ | $\mathbf{38.94}_{\pm0.47}$ | $\mathbf{70.18}_{\pm0.23}$ |
| | First-task HPO | $83.20_{\pm0.79}$ | $96.67_{\pm0.18}$ | $56.36_{\pm0.29}$ | $86.11_{\pm0.154}$ | $36.94_{\pm0.67}$ | $68.16_{\pm0.30}$ |
| | Current-task HPO | $\mathbf{83.99}_{\pm0.22}$ | $96.58_{\pm0.15}$ | $56.46_{\pm0.36}$ | $86.35_{\pm0.02}$ | $37.63_{\pm0.38}$ | $68.25_{\pm0.41}$ |
| | Seen-tasks HPO (Val) | $81.94_{\pm1.55}$ | $95.90_{\pm0.51}$ | $54.37_{\pm0.25}$ | $85.02_{\pm0.14}$ | $36.06_{\pm0.37}$ | $67.69_{\pm0.26}$ |
| | Seen-tasks HPO (Mem) | $81.61_{\pm0.15}$ | $96.40_{\pm0.13}$ | $53.76_{\pm0.21}$ | $84.56_{\pm0.31}$ | $32.37_{\pm0.34}$ | $64.37_{\pm0.47}$ |
| ESMER | End-of-training HPO | $80.73_{\pm0.15}$ | $96.50_{\pm0.01}$ | $56.16_{\pm0.54}$ | $88.69_{\pm0.35}$ | $\mathbf{47.33}_{\pm0.30}$ | $76.18_{\pm0.22}$ |
| | First-task HPO | $77.89_{\pm0.46}$ | $96.15_{\pm0.12}$ | $56.61_{\pm0.20}$ | $89.05_{\pm0.10}$ | $46.69_{\pm0.56}$ | $75.72_{\pm0.24}$ |
| | Current-task HPO | $81.69_{\pm0.25}$ | $96.03_{\pm0.05}$ | $55.11_{\pm0.13}$ | $88.96_{\pm0.08}$ | $45.20_{\pm0.53}$ | $74.93_{\pm0.29}$ |
| | Seen-tasks HPO (Val) | $81.29_{\pm0.03}$ | $96.46_{\pm0.06}$ | $53.81_{\pm0.44}$ | $87.26_{\pm0.13}$ | $44.82_{\pm0.16}$ | $74.27_{\pm0.11}$ |
| | Seen-tasks HPO (Mem) | $70.95_{\pm0.94}$ | $95.79_{\pm0.14}$ | $\mathbf{57.50}_{\pm0.14}$ | $89.27_{\pm0.16}$ | $44.26_{\pm0.20}$ | $74.54_{\pm0.31}$ |
| DER++ | End-of-training HPO | $84.40_{\pm0.94}$ | $95.75_{\pm0.33}$ | $56.04_{\pm3.67}$ | $83.13_{\pm2.69}$ | $\mathbf{39.89}_{\pm0.27}$ | $\mathbf{70.41}_{\pm0.17}$ |
| | First-task HPO | $85.22_{\pm0.08}$ | $96.14_{\pm0.10}$ | $55.20_{\pm0.78}$ | $81.68_{\pm0.66}$ | $35.98_{\pm0.63}$ | $65.86_{\pm0.37}$ |
| | Current-task HPO | $84.90_{\pm0.11}$ | $95.92_{\pm0.11}$ | $55.00_{\pm1.21}$ | $83.14_{\pm0.76}$ | $36.64_{\pm0.33}$ | $66.43_{\pm0.49}$ |
| | Seen-tasks HPO (Val) | $85.44_{\pm0.38}$ | $96.22_{\pm0.15}$ | $56.59_{\pm0.64}$ | $83.61_{\pm0.42}$ | $31.88_{\pm5.36}$ | $64.20_{\pm3.00}$ |
| | Seen-tasks HPO (Mem) | $82.18_{\pm0.26}$ | $94.75_{\pm0.28}$ | $56.94_{\pm0.66}$ | $83.08_{\pm0.21}$ | $33.54_{\pm0.13}$ | $63.68_{\pm0.17}$ |

chose to bold results in this way is to be able to draw attention to and reference observed effect sizes. We want to do this as if the observed effect sizes are small it suggests that no method performs much better than any other and hence that other factors become more important when selecting a HPO framework, e.g. compute cost. In Table 2 there are not many bolded numbers and for those that exist, the HPO framework which achieves it varies. This shows that no HPO framework performs consistently better than the rest. For instance, on CIFAR-100, no HPO framework improves accuracy over the other methods by more than +0.5% for all CL methods but ESMER in class-incremental learning. This suggest that for the split task setting there is no general advantage in using one HPO framework over another in terms of predictive performance.

In the heterogeneous task setting we also see that none of the HPO frameworks perform consistently better than the rest. The results for this setting are presented in Table 3 and we have again bolded the results which are better by +0.5% than any of the other HPO frameworks for a given CL method. Like the results for the split task setting, there are many columns for each CL method which have no bolded result and for the three which do the HPO framework which achieves it is different. Therefore, we conclude that in the heterogeneous task setting it is also the case that there is no one best HPO framework. We also note that the reason we look at the heterogeneous task setting is because we expected a greater benefit from adapting hyperparameters per task, given that unlike the split task setting each task is quite different. However, our

Table 3: Results of using different HPO frameworks for ER, iCaRL, ER-ACE, ESMER and DER++ on heterogeneous task benchmarks. We report mean average accuracy over three runs with their standard errors and bold the results which are greater by +0.5% accuracy than any other for that CL method. The table shows that the simplest and least computationally expensive method, first-task HPO, performs similarly to the other HPO frameworks and that no HPO framework is consistently better than the rest.

| CL Method | HPO Framework | Hetero-CIFAR-100 Class-IL. | Hetero-TinyImg Class-IL. |
|---|---|---|---|
| ER | End-of-training HPO | $50.41_{\pm 0.21}$ | $39.41_{\pm 0.57}$ |
| | First-task HPO | $50.33_{\pm 0.50}$ | $40.77_{\pm 0.34}$ |
| | Current-task HPO | $49.77_{\pm 0.21}$ | $40.65_{\pm 0.97}$ |
| | Seen-tasks HPO (Val) | $\mathbf{51.70}_{\pm 0.23}$ | $40.55_{\pm 0.22}$ |
| | Seen-tasks HPO (Mem) | $45.52_{\pm 0.41}$ | $\mathbf{44.62}_{\pm 0.18}$ |
| iCaRL | End-of-training HPO | $51.54_{\pm 0.38}$ | $37.17_{\pm 0.48}$ |
| | First-task HPO | $49.81_{\pm 0.10}$ | $37.47_{\pm 0.26}$ |
| | Current-task HPO | $51.34_{\pm 0.32}$ | $37.07_{\pm 0.07}$ |
| | Seen-tasks HPO (Val) | $48.15_{\pm 0.09}$ | $35.70_{\pm 0.23}$ |
| | Seen-tasks HPO (Mem) | $47.87_{\pm 0.15}$ | $35.27_{\pm 1.12}$ |
| ER-ACE | End-of-training HPO | $51.96_{\pm 0.60}$ | $\mathbf{45.47}_{\pm 0.42}$ |
| | First-task HPO | $51.37_{\pm 0.16}$ | $43.62_{\pm 1.09}$ |
| | Current-task HPO | $51.78_{\pm 0.30}$ | $43.87_{\pm 0.20}$ |
| | Seen-tasks HPO (Val) | $51.94_{\pm 0.12}$ | $43.15_{\pm 0.63}$ |
| | Seen-tasks HPO (Mem) | $48.15_{\pm 0.28}$ | $42.19_{\pm 0.84}$ |
| ESMER | End-of-training HPO | $50.54_{\pm 0.16}$ | $44.87_{\pm 0.26}$ |
| | First-task HPO | $50.43_{\pm 0.34}$ | $45.84_{\pm 0.50}$ |
| | Current-task HPO | $50.68_{\pm 0.31}$ | $44.50_{\pm 0.31}$ |
| | Seen-tasks HPO (Val) | $47.96_{\pm 0.61}$ | $42.18_{\pm 0.22}$ |
| | Seen-tasks HPO (Mem) | $50.56_{\pm 0.40}$ | $46.00_{\pm 0.43}$ |
| DER++ | End-of-training HPO | $54.12_{\pm 0.70}$ | $46.41_{\pm 0.77}$ |
| | First-task HPO | $54.87_{\pm 0.39}$ | $43.45_{\pm 3.55}$ |
| | Current-task HPO | $55.10_{\pm 0.52}$ | $45.95_{\pm 0.93}$ |
| | Seen-tasks HPO (Val) | $54.67_{\pm 0.57}$ | $46.51_{\pm 0.49}$ |
| | Seen-tasks HPO (Mem) | $49.06_{\pm 3.90}$ | $25.78_{\pm 7.40}$ |

results show that this is not the case and that it is possible to use the same hyperparameters across all the tasks and still perform well.

**Performance of first-task HPO** Our results show that all of the HPO frameworks tested perform similarly. Therefore, we conclude that first-task HPO is a good method to use as it is the most computationally efficient. We describe here in more detail its relative performance compared to the other HPO frameworks tested. In the split tasks setting, we see from Table 2, that for ER some of its results are bolded. Thus, first-task HPO sometimes achieves the best performance. Additionally, for the spilt task setting, there is an average performance difference from end-of-training HPO to first-task HPO of $-0.42\%$ in class-incremental learning and $-0.84\%$ in task-incremental learning. While, for the heterogeneous tasks setting there is an average performance difference from end-of-training HPO to first-task HPO of $-0.39\%$. End-of-training HPO is currently the most used HPO framework in CL but is unrealistic and computationally expensive. Hence, our results indicate that by using first-task framework it is possible to perform realistic HPO for much less computation with only a small expected cost to performance.

One of the potential reasons that the performance is similar between HPO frameworks is that there is little variation between the performance of different hyperparameter configurations. To see whether this

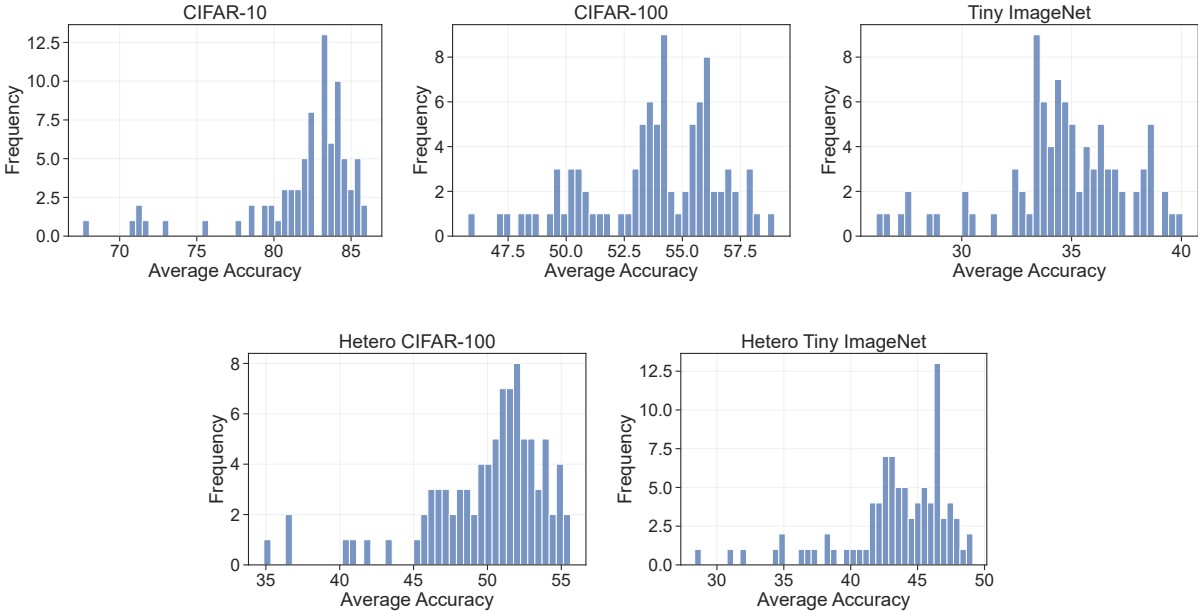

Figure 3: Histograms of the validation accuracy at the end of training for each hyperparameter setting searched over for DER++. We look at standard CL benchmarks and heterogeneous task benchmarks, which are identified by having a 'Hetero' in their name. The histograms show that different hyperparameter settings give a varying range of performances and only a few achieve near to the top performance.

is the case, we have plotted in Figure 3 histograms of the performance of using different fixed HPO configurations for DER++. The histograms show that hyperparameter configurations achieve a wide range of average accuracies. Therefore, the performance of different HPO configurations is *not* the reason why the HPO frameworks have similar results. Additionally, in Appendix B, we examine whether using default hyperparameters performs as well as selecting hyperparameters using HPO. We found that using default hyperparameters in most cases performed worse than using a HPO framework. Hence, our results suggest that HPO is necessary but that out of the HPO frameworks tested there is no one best performing method.

## 6 Conclusions

In this paper we have benchmarked several hyperparameter optimisation (HPO) frameworks for CL which are more realistic than the currently commonly used end-of-training HPO framework. We benchmarked both fixed HPO frameworks, which fix the hyperparameters throughout training, and dynamic HPO frameworks that continually adapt the hyperparameters. Our results show that all the HPO frameworks achieve similar performances and none consistently outperforms the others. Because of this, we recommend that the preferred HPO framework for future work on standard continual learning benchmarks should be the much more computationally efficient *first-task* HPO. However, this is just our recommendation and it could be that for a given benchmark and/or CL method that another realistic HPO method should be preferred. For future work, we note that we have only looked at empirical evidence and so it would also be useful to see what theoretical results are achievable as well.

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

## A    Additional experimental details

While we have aimed to include all the main experimental details in the main paper there are a few others to mention here. First, we mostly follow the experimental setup of Buzzega et al. (2020) and Boschini et al. (2022) and use the Mammoth library produced by those works as the base of our code. Second, we use as our optimiser SGD with no momentum or weigh decay, as is done in other works (Aljundi et al., 2019b; Buzzega et al., 2020; Chaudhry et al., 2019a; Lee & Storkey, 2023). Third, in the heterogeneous tasks setting we look at tasks sequences where each task in order has the following number of classes associated with it $[9, 2, 7, 3, 4, 9, 8, 3, 3, 7, 4, 4, 5, 9, 4, 5, 2, 8, 2, 2]$ and all the data of a class is contained in the task associated with it. For Tiny ImageNet we only use the first 100 classes in the heterogeneous tasks setting to reduce runtime and to make it more comparable to CIFAR-100 in that setting. In the heterogeneous tasks setting each task has a variable amount of data. For example, using CIFAR-100, the first task contains nine classes and so it will contain in total 4500 examples (500 examples per task) while the second task contains two classes so will only contain 1000 examples. Also, as in each task the learner needs to discriminate between a varying number of classes the difficultly should vary between tasks. Additionally, in the heterogeneous tasks setting we only look at class-incremental learning.

We record here the hyperparameter grid that we sample over when performing HPO. We look at learning rates in the set $\{0.2, 0.15, 0.1, 0.075, 0.05, 0.03, 0.01, 0.0075, 0.005, 0.0025\}$. For DER++, we perform HPO over both regularisation coefficients where we sample $\alpha$ in the set $\{0.2, 0.5, 1.0\}$ and $\beta$ in the set $\{0.2, 0.5, 1.0\}$. For ESMER, we perform HPO over the loss margin coefficient where we sample over the set $\{1.5, 1.2, 1.0\}$. We sample all possible combinations of learning rates and regularisation coefficients in each of our HPO frameworks. This grid contains the ones used in the popular works Buzzega et al. (2020), Boschini et al. (2022) and Sarfraz et al. (2023), where we add additional learning rate settings and, for some datasets, regularisation coefficients settings. We note here that while we use grid search in this paper to align with common practice in CL (Buzzega et al., 2020; Delange et al., 2021), any hyperparameter sampling/selecting method can be used with each of the HPO frameworks looked at. For example, tree-structured Parzen estimators are a common Bayesian HPO method to sample hyperparameter configurations for neural networks (Bergstra et al., 2011). Additionally, Gaussian process based HPO methods are also commonly used (Snoek et al., 2012) and have been looked at in settings related to online learning (Hellan et al., 2023).

## B    Experiments using default hyperparameter values

To test whether HPO is needed in CL and if instead using default hyperparameters is sufficient, we perform experiments using default hyperparameters. The experimental setup is the same as the main paper and we use for the default learning rate the default given by PyTorch, 0.001, and use 1.0 as the default for regularisation coefficients. The results are presented in Tables 4 and 5. The tables show that using default hyperparameters leads to worse performance than using HPO. Additionally, for some dataset and CL method combinations the default hyperparameters perform very badly showing the need to adapt hyperparameters to the dataset and CL method used.

Table 4: Comparison of using default hyperparameters versus using a HPO framework on standard CL benchmarks, where we only present the most common HPO framework (End-of-training HPO) and the most efficient (First-task HPO) for readability. We report mean average accuracies over three runs with their standard errors. The table shows that using default HPs leads to worse performance than using HPO for standard CL benchmarks.

| | | CIFAR-10 | | CIFAR-100 | | TinyImageNet | |
|---|---|---|---|---|---|---|---|
| CL Method | HPO Framework | Class-IL. | Task-IL. | Class-IL. | Task-IL. | Class-IL. | Task-IL. |
| ER | End-of-training HPO | $83.55_{\pm 0.44}$ | $97.18_{\pm 0.14}$ | $51.03_{\pm 0.43}$ | $85.68_{\pm 0.29}$ | $28.01_{\pm 0.09}$ | $68.17_{\pm 0.06}$ |
| | First-task HPO | $84.38_{\pm 0.45}$ | $96.82_{\pm 0.17}$ | $49.61_{\pm 0.34}$ | $84.97_{\pm 0.19}$ | $28.51_{\pm 0.18}$ | $68.72_{\pm 0.13}$ |
| | Default HPs | $74.60_{\pm 0.79}$ | $94.53_{\pm 0.13}$ | $35.39_{\pm 0.36}$ | $72.83_{\pm 0.24}$ | $16.27_{\pm 0.20}$ | $50.99_{\pm 0.41}$ |
| iCaRL | End-of-training HPO | $77.79_{\pm 0.23}$ | $98.52_{\pm 0.03}$ | $54.30_{\pm 0.36}$ | $85.74_{\pm 0.45}$ | $37.09_{\pm 0.27}$ | $70.37_{\pm 0.36}$ |
| | First-task HPO | $77.83_{\pm 0.22}$ | $95.31_{\pm 0.12}$ | $52.56_{\pm 0.10}$ | $84.60_{\pm 0.09}$ | $36.42_{\pm 0.22}$ | $70.11_{\pm 0.13}$ |
| | Default HPs | $68.34_{\pm 0.49}$ | $92.98_{\pm 0.21}$ | $11.54_{\pm 0.25}$ | $41.66_{\pm 0.54}$ | $5.30_{\pm 0.03}$ | $23.97_{\pm 0.10}$ |
| ER-ACE | End-of-training HPO | $82.34_{\pm 0.30}$ | $96.74_{\pm 0.01}$ | $55.58_{\pm 0.39}$ | $85.73_{\pm 0.09}$ | $38.94_{\pm 0.47}$ | $70.18_{\pm 0.23}$ |
| | First-task HPO | $83.20_{\pm 0.79}$ | $96.67_{\pm 0.18}$ | $56.36_{\pm 0.29}$ | $86.11_{\pm 0.154}$ | $36.94_{\pm 0.67}$ | $68.16_{\pm 0.30}$ |
| | Default HPs | $75.46_{\pm 0.21}$ | $94.71_{\pm 0.06}$ | $42.65_{\pm 0.57}$ | $76.28_{\pm 0.19}$ | $25.84_{\pm 0.26}$ | $56.25_{\pm 0.13}$ |
| ESMER | End-of-training HPO | $80.73_{\pm 0.15}$ | $96.50_{\pm 0.01}$ | $56.16_{\pm 0.54}$ | $88.69_{\pm 0.35}$ | $47.33_{\pm 0.30}$ | $76.18_{\pm 0.22}$ |
| | First-task HPO | $77.89_{\pm 0.46}$ | $96.15_{\pm 0.12}$ | $56.61_{\pm 0.20}$ | $89.05_{\pm 0.10}$ | $46.69_{\pm 0.56}$ | $75.72_{\pm 0.24}$ |
| | Default HPs | $68.86_{\pm 1.06}$ | $93.54_{\pm 0.20}$ | $42.94_{\pm 0.61}$ | $79.64_{\pm 0.36}$ | $33.11_{\pm 0.39}$ | $63.15_{\pm 0.17}$ |
| DER++ | End-of-training HPO | $84.40_{\pm 0.94}$ | $95.75_{\pm 0.33}$ | $56.04_{\pm 3.67}$ | $83.13_{\pm 2.69}$ | $39.89_{\pm 0.27}$ | $70.41_{\pm 0.17}$ |
| | First-task HPO | $85.22_{\pm 0.08}$ | $96.14_{\pm 0.10}$ | $55.20_{\pm 0.78}$ | $81.68_{\pm 0.66}$ | $35.98_{\pm 0.63}$ | $65.86_{\pm 0.37}$ |
| | Default HPs | $77.59_{\pm 0.45}$ | $93.83_{\pm 0.40}$ | $46.11_{\pm 1.16}$ | $78.14_{\pm 1.28}$ | $25.66_{\pm 0.16}$ | $59.14_{\pm 0.51}$ |

Table 5: Comparison of using default hyperparameters versus using a HPO framework on heterogeneous task benchmarks, where we only present the most common HPO framework (End-of-training HPO) and the most efficient (First-task HPO) for readability. We report mean average accuracies over three runs with their standard errors. The table shows that using default HPs leads to worse performance than using HPO for heterogeneous task benchmarks.

| | | Hetero-CIFAR-100 | Hetero-TinyImg |
|---|---|---|---|
| CL Method | HPO Framework | Class-IL. | Class-IL. |
| ER | End-of-training HPO | $50.41_{\pm 0.21}$ | $39.41_{\pm 0.57}$ |
| | First-task HPO | $50.33_{\pm 0.50}$ | $40.77_{\pm 0.34}$ |
| | Default HPs | $33.76_{\pm 0.78}$ | $26.88_{\pm 0.45}$ |
| iCaRL | End-of-training HPO | $51.54_{\pm 0.38}$ | $37.17_{\pm 0.48}$ |
| | First-task HPO | $49.81_{\pm 0.10}$ | $37.47_{\pm 0.26}$ |
| | Default HPs | $12.23_{\pm 0.19}$ | $10.6_{\pm 0.26}$ |
| ER-ACE | End-of-training HPO | $51.96_{\pm 0.60}$ | $45.47_{\pm 0.42}$ |
| | First-task HPO | $51.37_{\pm 0.16}$ | $43.62_{\pm 1.09}$ |
| | Default HPs | $38.11_{\pm 0.80}$ | $32.37_{\pm 0.53}$ |
| ESMER | End-of-training HPO | $50.54_{\pm 0.16}$ | $44.87_{\pm 0.26}$ |
| | First-task HPO | $50.43_{\pm 0.34}$ | $45.84_{\pm 0.50}$ |
| | Default HPs | $37.92_{\pm 0.30}$ | $34.22_{\pm 0.41}$ |
| DER++ | End-of-training HPO | $54.12_{\pm 0.70}$ | $46.41_{\pm 0.77}$ |
| | First-task HPO | $54.87_{\pm 0.39}$ | $43.45_{\pm 3.55}$ |
| | Default HPs | $44.43_{\pm 0.51}$ | $30.21_{\pm 1.53}$ |

