# OpenReview forum: "Hyperparameter Selection in Continual Learning"
_TMLR — Rejected by TMLR_

### Review · Reviewer_V1Xx · 2024-05-12

**Summary Of Contributions:**

This paper considers the problem of hyperparameter selection in continual learning. It benchmarks several common methods on CIFAR10, CIFAR100, and TinyImageNet with a variety of continual learning methods. The authors conclude that the methods all perform similarly and recommend that practitioners use first-task HPO since it's the least computationally expensive.

**Audience:**

Yes

**Broader Impact Concerns:**

The authors do not bring up broader impacts. Since the work considers a general-purpose optimization technique, I think that is fine.

**Claims And Evidence:**

Yes

**Requested Changes:**

### Critical ###

Modify the analysis of the experiments to consider the standard errors.

### Strengthen ###

It would greatly strengthen the work to include a wider range of tasks in the empirical study. However, since most continual learning benchmarks are based on image classification, I do not consider this to be a critical issue.

**Strengths And Weaknesses:**

### Strengths ###
1. The paper is clearly written and straightforward to understand. Each of the benchmarked methods is explained in detail.
2. The paper provides a clear recommendation for practitioners.
3. The paper discusses the computational complexity of the methods in addition to experiments.

### Weaknesses ###
1. Experiments are done only on image classification benchmarks. It would greatly strengthen the work to conduct experiments on a greater range of tasks, for example CLOC (https://arxiv.org/pdf/2108.09020) or CoRe50 (https://vlomonaco.github.io/core50/benchmarks.html)
2. Comparing the results of different methods is done using a 0.5% threshold. To be rigorous, the comparisons should be done utilizing the standard errors via statistical tests.

---

> ### Author Response · Authors · 2024-05-30
> **Author Response**
>
> Dear reviewer, thank you for your comments and questions, we provide answers to them below.
>
> ****1. Comparing the results of different methods is done using a 0.5\% threshold. To be rigorous, the comparisons should be done utilizing the standard errors via statistical tests.****
>
> Given your comment we have computed statistical tests for our experiments. We used pairwise Welch's t-tests and Bonferroni corrections to test for a given method/dataset/setting combination whether one of the HPO frameworks performed statistically significantly better than the rest. We found that at the 95\% level none of the HPO frameworks are statistically significantly better for any method/dataset/setting combination.
>
> Additionally, we would like to mention here why we chose the presentation choice of bolding results which are +0.5\% greater than any other for a given method/dataset/setting combination. This is because we wanted to examine effect size. The reason we think effect size is a good thing to look at when analysing the results is that we want to argue that no HPO framework performs that much better than another and hence that other factors like compute should be used to select a HPO framework. In other words while the true performance of two HPO methods may be different, if it differs only in the 10th decimal place, i.e. a small effect size, this does not matter that much and so other factors like compute should be then used for selection. Hence, by bolding results that are +0.5\% greater than any other we can draw attention to and reference observed effect sizes in the paper and so clearly argue and evidence our claim. We thank you for this comment and understand that we did not make clear why we bold results which are +0.5\% greater than any other, so have edited the text accordingly.
>
> ****2. It would greatly strengthen the work to include a wider range of tasks in the empirical study. However, since most continual learning benchmarks are based on image classification, I do not consider this to be a critical issue.****
>
> As suggested by you, we have run experiments for CORe50 and have presented them in the general comment we have posted.
>
> ****
>
> We hope that both of our answers to your requested changes are satisfactory and thank you for making them as we think they have strengthened the work.

---

### Review · Reviewer_NGTb · 2024-05-17

**Summary Of Contributions:**

This paper studies the problem of which hyperparameter optimization (HPO) framework is better for continual learning. Several frameworks are tested on a few benchmarks to compare their performances. The authors show their performances are similar and thus the best one is the most efficient and realistic one, which is first-task HPO.

**Audience:**

Yes

**Claims And Evidence:**

Yes

**Requested Changes:**

The authors might want to consider studying more interesting and challenging continual learning problems to improve the significance of the paper.

**Strengths And Weaknesses:**

**Strengths:**
- Paper is written clearly and easy to follow
- Overall presentation has good quality

**Weaknesses:**
- Novelty: the paper mainly studies existing frameworks and the novelty seems limited.
- Significance: the authors argue all HPO frameworks show the same performance. While this can be an interesting result: on certain benchmarks HPO framework choices simply do not matter too much, but at the same time, I think this indicates the benchmarks chosen in this paper are simply not that interesting for continual learning. There are certainly other more challenging and non-static continual learning problems where a better HPO strategy is required. Studying these basic benchmarks can be interesting, but the significance of the contribution will be limited.

---

> ### Author Response · Authors · 2024-05-30
> **Author Response**
>
> Dear reviewer, we thank you for your comments. We provide an answer to your requested change below.
>
> ****1. The authors might want to consider studying more interesting and challenging continual learning problems to improve the significance of the paper.****
>
> We have added experiments on the more challenging CORe50 dataset, suggested by reviewer V1Xx, which are discussed in the general comment we posted.
>
> We hope that this has satisfied your requested change.

---

### Review · Reviewer_8Uuv · 2024-05-24

**Summary Of Contributions:**

The paper compares different algorithms for continual learning in terms of computational complexity and experimental performance on benchmark datasets.

**Audience:**

Yes

**Claims And Evidence:**

No

**Requested Changes:**

I would like to see the following before accepting:
- Clarification of exactly why end-of-training HPO is unrealistic.
- Connection to some practical application outside of the continual learning benchmarking experiments shown here.

**Strengths And Weaknesses:**

Strengths:
- The presentation is mostly clear.
- The experiments are well done.

Weaknesses:
- The authors claim that the standard method for tuning hyperparameters in continual learning (end-of-training HPO) is unrealistic because the model should only be able to make decisions based on streaming tasks (without access to the full history). However, it is not clear what constraints make this unrealistic. Presumably, the main constraint is the space complexity; the algorithms here require enough space to store the dataset for a single task but the constraint prevents storing the data for all tasks. In this scenario it would still be realistic to use end-of-training HPO by training a set of models with different hyperparameter combinations in parallel and doing model selection at the end. Thus, end-of-training HPO is only unrealistic due space complexity if one needs to scale the total number of hyperparameter combinations to the point where the storage required for all the models exceeds the space of the dataset for a task.
- It is unclear to me whether there are practical applications for this work. It would be useful to have real examples or at least realistic scenarios where this might be applied.
- There is a lack of theoretical motivation. Why should we expect one method to perform better than another? What happens in the limiting cases? What properties of the data would affect the choice of algorithm?

---

> ### Author Response · Authors · 2024-05-30
> **Author Response**
>
> Dear reviewer, thank you for your review, we provide answers to your comments and questions below.
>
>
> ****1. Clarification of exactly why end-of-training HPO is unrealistic.****
>
> In this work we look at HPO for CL, which means that we inherit all the assumptions and resource constraints of the base CL setting and methods. One of the main constraints, as you point out, is on the amount of storage a learner uses. Because of this, in the standard CL setup it is not possible to store a large amount of additional data or networks in memory. Hence, it is not possible to store lots of networks, one for each hyperparameter configuration, to perform HPO. Additionally, for the HPO framework you outline, the learner would have to store the validation data for each previous task and not train on it when observing that task, which makes it slightly different to end-of-training HPO. To make the amount of extra memory usage more clear, using DER++ with the HPO method you outline would require roughly the same amount of memory as storing all of CIFAR10 or CIFAR100 13 times over. Last, we would like to point out that our recommended method---first-task HPO---has another advantage over end-of-task HPO, being much more computationally efficient and with negligible space overhead. We thank the reviewer for the question as we now understand that we did not make clear fully why end-of-training HPO is unrealistic in the paper and so have edited it accordingly.
>
> ****2. It is unclear to me whether there are practical applications for this work.****
>
> We see the main practical impact of the paper to be informing best practice for HPO in CL research. This is because we look at the main standard benchmarks used in CL and show that it is possible to perform HPO on these benchmarks realistically and at a fraction of the computational cost of what is currently done (i.e. end-of-task HPO). Therefore, this work will certainly impact our own future work in CL by showing how to reduce the computational burden and energy costs and hopefully other CL researchers' work as well. More generally, CL as a whole is not really at a stage where it can be used widely and easily in real-world settings. So, with work like ours we can improve the speed at which new CL methods can be developed and evaluated, helping drive CL towards real-world use. Additionally, our results show that it is possible to perform HPO for CL well in a realistic manner, helping make CL more practical.
>
> ***
>
> Thank you again for your review which we believe will improve the quality of the paper and we hope especially that we have explained clearly why end-of-training HPO is unrealistic.

---

> > ### Comment · Reviewer_8Uuv · 2024-05-30
> >
> > Thank you for the responses.
> >
> > 1) Regarding the complexity of end-of-training HPO, I agree with your response but I would like see the space complexity made explicit in the paper. I believe this will help readers judge whether the savings justify the potential costs.
> >
> > 2) According to the authors, the primary use of the proposed method is to speed up experiments on CL algorithms. For this application, space complexity is not a bottleneck, so the real advantage is some computation time savings. The authors claim that first-task HPO is just as good as end-of-training HPO for this application, but I find the argument unconvincing. There is no theoretical argument for this claim, and these experiments are limited. As mentioned by the other reviewers, the tasks presented are relatively simple. The conclusions will not necessarily generalize to more complicated tasks. Similarly, the conclusion will not necessarily generalize to new CL algorithms. Ultimately, I would not trust experimental results comparing CL algorithms if they relied on first-task HPO.
> >
> > I think the burden of proof lies with the authors. While first-task HPO not is uniformly worse than end-of-training HPO, it is indeed worse on some tasks (e.g. the first experiment presented in the response). One possible explanation is that first-task HPO works equally well for some simple benchmarks but does worse on more complex benchmarks. I think the authors need to limit their scope of their claim to a subset of problems for which they can provide evidence.

---

> > > ### Author Response · Authors · 2024-06-04
> > > **Author Response**
> > >
> > > Thank you for your response and additional comments. We provided answers to them below.
> > >
> > > ****1. Regarding the complexity of end-of-training HPO, I agree with your response but I would like see the space complexity made explicit in the paper. I believe this will help readers judge whether the savings justify the potential costs.****
> > >
> > > We agree with you and have added a discussion of the space complexity of end-of-training HPO to Section 4.
> > >
> > > ****2. End-of-training HPO being unrealistic.****
> > >
> > > Any HPO framework for continual learning must adhere to the standard constraints of the CL setting, including limited storage. Hence, in all our experiments, memory is a bottleneck. This is why it is not possible to perform, a variant of, end-of-training HPO by training dozens of models in parallel. It may also be unpractical due to limited compute resources. Alternatively, it is not possible to perform end-of-training HPO sequentially since it would require re-training the network many times over the data stream, which should only be seen once and cannot be stored fully in memory. Therefore, both of these versions of end-of-training HPO are unrealistic and when used in research could mislead what the actual performance of a CL method is. This is the key motivation of our paper. We must consider alternative realistic HPO frameworks and in our experiments, we find that all perform similarly. The computational speed-up of first-task HPO is an added benefit and is the reason we recommend its use, for the common CL benchmarks used in our experiments.
> > >
> > > ****3. Scope of empirical claims.****
> > >
> > > We agree that, like any solely-empirical benchmarking paper, we can only make claims about the performance of a HPO framework for the datasets, settings and CL methods tested in our experiments. So, given your comment, we have looked through the paper and ensured that all our claims are limited to this scope. Also, we note that this is the reason we chose the datasets/settings looked at in our experiments as these are the most commonly used benchmarks in CL, to the best of our knowledge. Therefore, by using them we can test what HPO framework performs well on the benchmarks current CL practitioners mostly use. Last, we agree with the reviewer that theoretical work in this area would be great but it is non-trivial and is out of scope for this empirical benchmarking paper. Therefore, we have added the suggestion of theoretical results in the discussion of future work in the paper.

---

> > > > ### Comment · Reviewer_8Uuv · 2024-06-11
> > > >
> > > > Thank you for adding the clarification on the space complexity of End of Training HPO to Section 4.
> > > >
> > > > I still disagree with the following claims of the paper:
> > > > 1. The authors claim that end-of-training HPO is unrealistic due to computational constraints. The time complexity is advantage is clear. The space complexity is less clear, but is the basis for the "unrealistic" claim. This constraint on space complexity is only relevant in a very narrow setting that is still not presented clearly in the paper. The authors generalize too much by claiming that end of training HPO is "unrealistic" without clearly laying out their argument.
> > > > 1. The authors "see the main practical impact of the paper to be informing best practice for HPO in CL research." However, CL researchers are extremely unlikely to be constrained by storage space. So the only advantage here is some convenience for CL researchers.
> > > > 1. The authors claim that first task HPO is as good as end of training HPO. I don't see sufficient evidence to draw this conclusion.
> > > > 1. The authors claim that the evidence for first task HPO being as good as end of training HPO is strong enough for researchers to prefer first task HPO in CL research. From the conclusion, "We recommend that the preferred HPO framework for future work on standard continual learning benchmarks should be the much more computationally efficient first-task HPO." Given that the evidence for first task being equivalent to end of training HPO is weak, and the only advantage is convenience, I think this conclusion is not justified.

---

> ### Author Response · Authors · 2024-06-13
> **Author Response**
>
> Thank you for your additional comments, we answer them below.
>
> ****1. The memory constraints of CL****
>
> Given all your comments, it seems that the main problem you have is about the storage space usage in CL, where you believe that it is not a constraint. This is not true, in most work on CL storage is the main constraint/bottleneck, as described in these CL survey papers: De Lange et al. (2021) and Wang et al. (2024). However, we understand that we did not make this clear in the paper and so have added a discussion of it to the preliminaries/related work section. Additionally, we would like to say that as mentioned before, the memory usage of "paralyzing" end-of-training HPO would exceed the memory required to store any of the datasets we look at. Therefore, if a learner had that much memory it could trivially solve standard CL by storing all the data and performing offline learning, which achieves much better performance than any of the CL methods looked at (for a further discussion of this point see Prabhu et al. (2020)).
>
> ****2. Performance of first task HPO****
>
> We do not claim that first-task HPO is just good as end-of-training HPO. Instead, in the paper we give a detailed analysis of the performance difference between first-task HPO and end-of-training HPO in the results section (in the paragraph that starts with the bold text "Performance of first-task HPO"). In it we state that the performance of first-task HPO is slightly worse on average, -0.42\% for class-incremental learning and -0.84\% for task-incremental learning. However, as end-of-training HPO is unrealistic it cannot be used in practice so you must use a framework like first-task or current-task HPO.
>
>
> ****
> M. De Lange, et al. A continual learning survey: Defying forgetting in classification tasks. IEEE transactions on pattern analysis and machine intelligence 44.7 (2021): 3366-3385.
>
> L. Wang, et al. A comprehensive survey of continual learning: Theory, method and application. IEEE Transactions on Pattern Analysis and Machine Intelligence (2024).
>
> A. Prabhu, et al. Gdumb: A simple approach that questions our progress in continual learning. ECCV. 2020.

---

### Author Response · Authors · 2024-05-30
**Results on CORe50**

Reviewers NGTb and V1Xx felt that the paper would benefit by including a more challenging continual learning benchmark. Because of this feedback, we have now included results on the more diverse CORe50 (NC) benchmark, suggested by reviewer V1Xx. The results for ER, iCarL and ER-ACE are presented in the table below and show the same outcome as the rest of our experiments, in that no HPO framework performs consistently better than the rest. Additionally, we have started running the experiments for DER++ and ESMER but given the significant computational cost of running many HPO sweeps these will not finish for over a week.


| CL Method | HPO Framework        | Class-IL.          | Task-IL.           |
|-----------|:----------------------|--------------------:|--------------------|
| **ER**        | End-of-training HPO  | $39.90\_{ \pm 0.30}$ | $58.53 \_{ \pm 0.33}$ |
|           | First-task HPO       | $36.39 \_{ \pm 1.34}$ | $53.34 \_{ \pm 1.21}$ |
|           | Current-task HPO     | $37.20 \_{ \pm 1.52}$ | $56.10 \_{ \pm 2.23}$ |
|           | Seen-tasks HPO (Val) | $36.49 \_{ \pm 0.91}$ | $52.36 \_{ \pm 1.12}$ |
|           | Seen-tasks HPO (Mem) | $36.72 \_{ \pm 0.52}$ | $54.90 \_{ \pm 1.13}$ |
| **iCaRL**     | End-of-training HPO  | $36.01 \_{ \pm 0.70}$ | $54.42 \_{ \pm 0.72}$ |
|           | First-task HPO       | $36.63 \_{ \pm 0.36}$ | $55.76 \_{ \pm 0.42}$ |
|           | Current-task HPO     | $36.37 \_{ \pm 1.36}$ | $55.14 \_{ \pm 1.88}$ |
|           | Seen-tasks HPO (Val) | $37.01 \_{ \pm 0.57}$ | $55.60 \_{ \pm 0.39}$ |
|           | Seen-tasks HPO (Mem) | $34.79 \_{ \pm 0.49}$ | $54.10 \_{ \pm 0.32}$ |
| **ER-ACE**    | End-of-training HPO  | $41.33 \_{ \pm 1.67}$ | $59.25 \_{ \pm 1.65}$ |
|           | First-task HPO       | $39.85 \_{ \pm 0.26}$ | $58.56 \_{ \pm 0.14}$ |
|           | Current-task HPO     | $44.02 \_{ \pm 1.30}$ | $62.05 \_{ \pm 1.12}$ |
|           | Seen-tasks HPO (Val) | $37.58 \_{ \pm 0.13}$ | $55.69 \_{ \pm 0.34}$ |
|           | Seen-tasks HPO (Mem) | $35.58 \_{ \pm 1.31}$ | $54.02 \_{ \pm 1.62}$ |

---

> ### Comment · Reviewer_V1Xx · 2024-06-11
> **Thanks for the additional experiments**
>
> Based on these results, the argument that the performance of First-task HPO is comparable to the other frameworks is less strong (see ER and ER-ACE). Therefore, I suspect that the story is quite nuanced, depending on the algorithms and benchmarks studied.

---

> > ### Author Response · Authors · 2024-06-13
> > **Author Response**
> >
> > Thank you for your response. If our understanding is correct you are saying that which HPO framework performs best depends on the CL method and benchmark used, which we agree with. This is a key claim of the paper, no HPO framework performs across the board better than the rest. Therefore, as we cannot tell a priori what realistic HPO framework performs the best for a new benchmark/CL-method, which one should a CL practitioner use? We suggest first-task HPO as it realistic and has the added benefit of being much more computationally efficient then the other frameworks (also for some dataset/method combinations first-task performs the best). We would also like to mention that the main point of the paper is just to benchmark realistic HPO frameworks to understand how they compare, which to the best of our knowledge has not been done before. Given our benchmarking a reader can listen to our suggestions or draw their own conclusions of what realistic HPO framework to use for a given benchmark/CL-method. We have added this point to the paper.

---

### Decision · Action_Editor_EDtn · 2024-07-02

**Recommendation:** Reject

**Comment:**

Dear authors,

Thank you for your submission to TMLR. While realistic hyperparameter optimization for continual learning is relevant to the TMLR audience, the reviewers are concerned about the strength of the claims and empirical evidence, especially after the additional results during the rebuttal.

(According to the TMLR acceptance criteria, any concerns about the submission's novelty or significance are irrelevant and, thus, are ignored in this decision.)

While the authors tried to address these concerns during the rebuttal, the reviewers were not fully satisfied with the responses, and the reviewers unanimously do not recommend accepting the submission in its current form.

Therefore, I recommend rejecting this submission. However, I strongly encourage the authors to consider the feedback and resubmit the paper after a major revision.

This major revision might involve broadening the scope of evaluation and providing clearer arguments for why first-task HPO is expected to perform well across settings—or reducing the scope: the reviewers seem to have focused their concerns around whether end-of-training HPO is unrealistic and whether first-task HPO makes for a good substitute while they generally commended the paper quality and experiment settings.

I want to thank the reviewers and the authors for the engaged discussions during the rebuttal and the extra experiments. I'm looking forward to the re-submission.

My best wishes

**Audience:**

Hyperparameter optimization for continual learning is of interest to at least some portion of TMLR's audience. Improving the efficiency and practicality of HPO methods for continual learning is a relevant problem for real-world settings with resource constraints.

**Claims And Evidence:**

The claims made in the submission are not fully supported by clear and convincing evidence: while the authors present results on several continual learning benchmarks, the reviewers raise concerns that the conclusions may only apply to the limited settings and benchmarks evaluated. Additional experiments on CORe50 presented in the rebuttal weaken the claim that first-task HPO performs comparably to other HPO frameworks. More theoretical justification and empirical evidence on a wider range of tasks would be needed to substantiate the claim that first-task HPO is the preferred choice. The authors have updated the paper throughout the rebuttal process, but the reviewers contend that the claims are not sufficiently evidenced.

**Resubmission Of Major Revision:**

The authors may consider submitting a major revision at a later time.